# The T cell receptor β chain repertoire of tumor infiltrating lymphocytes improves neoantigen prediction and prioritization

Thi Mong Quynh Pham[1], Thanh Nhan Nguyen[1], Bui Que Tran Nguyen[1], Thi Phuong Diem Tran[1], Nguyen My Diem Pham[1], Hoang Thien Phuc Nguyen[1], Thi Kim Cuong Ho[1], Dinh Viet Linh Nguyen[1], Huu Thinh Nguyen[2], Duc Huy Tran[2], Thanh Sang Tran[2], Truong Vinh Ngoc Pham[2], Minh Triet Le[2], Thi Tuong Vy Nguyen[1], Minh-Duy Phan[1], Hoa Giang[1], Hoai-Nghia Nguyen[1]*, Le Son Tran[1]*

[1]Medical Genetics Institute, Ho Chi Minh City, Viet Nam; [2]University Medical Center Ho Chi Minh City, Ho Chi Minh City, Viet Nam

## eLife assessment

The study presents a potentially **valuable** approach by combining two measurements (pHLA binding and pHLA-TCR binding) to improve predictions of which mutations in colorectal cancer are likely to be presented to and recognised by the immune system. While this approach is promising, the evidence supporting the primary claim remains somewhat **incomplete**. The experimental validation of the computational predictions with actual immune responses is still limited, despite the increase in sample size from 4 to 8 in this revision.

**\*For correspondence:**
nhnghia81@gmail.com (H-NN);
leson1808@gmail.com (LSonT)

**Abstract** In the realm of cancer immunotherapy, the meticulous selection of neoantigens plays a fundamental role in enhancing personalized treatments. Traditionally, this selection process has heavily relied on predicting the binding of peptides to human leukocyte antigens (pHLA). Nevertheless, this approach often overlooks the dynamic interaction between tumor cells and the immune system. In response to this limitation, we have developed an innovative prediction algorithm rooted in machine learning, integrating T cell receptor β chain (TCRβ) profiling data from colorectal cancer (CRC) patients for a more precise neoantigen prioritization. TCRβ sequencing was conducted to profile the TCR repertoire of tumor-infiltrating lymphocytes (TILs) from 28 CRC patients. The data unveiled both intra-tumor and inter-patient heterogeneity in the TCRβ repertoires of CRC patients, likely resulting from the stochastic utilization of V and J segments in response to neoantigens. Our novel combined model integrates pHLA binding information with pHLA-TCR binding to prioritize neoantigens, resulting in heightened specificity and sensitivity compared to models using individual features alone. The efficacy of our proposed model was corroborated through ELISpot assays on long peptides, performed on four CRC patients. These assays demonstrated that neoantigen candidates prioritized by our combined model outperformed predictions made by the established tool NetMHCpan. This comprehensive assessment underscores the significance of integrating pHLA binding with pHLA-TCR binding analysis for more effective immunotherapeutic strategies.

## Introduction

In metastatic CRC patients, immunotherapy has fulfilled the promise of improving survival rate (***Ganesh et al., 2019***). Immune checkpoint inhibitors (ICIs), which block negative regulatory pathways in T-cell

activation, have been approved by the US Food and Drug Administration (FDA) for the treatment of deficient mismatch repair (dMMR) or high microsatellite instability (MSI-H) CRC patients (*Dudley et al., 2016*; *Le et al., 2015*; *Overman et al., 2017*). However, there is an urgent need for alternative immunotherapy strategies for metastatic CRC patients, as patients with proficient mismatch repair (pMMR) or microsatellite stability (MSS) have not shown significant responses to immune checkpoint inhibitors (*Dudley et al., 2016*; *Le et al., 2017*).

Neoantigens (neopeptides) have emerged as potential targets for personalized cancer immuno-therapy, including CRC (*Yu et al., 2022*; *Kim et al., 2020*). Neoantigens are peptides that result from somatic mutations and can be displayed by class I human leukocyte antigen (HLA-I) molecules on the surface of cancer cells, thereby activating immune-mediated tumor killing (*Blass and Ott, 2021*). Recent studies have demonstrated that the presence of neoantigens is associated with better responses to immune checkpoint inhibitor (ICI) therapy in CRC patients (*Miao et al., 2018*; *Yarchoan et al., 2017*). A high neoantigen burden has been linked to improved overall survival and progression-free survival in patients with various solid tumors, including CRC (*Miao et al., 2018*; *Yarchoan et al., 2017*). Therefore, neoantigen-based immunotherapies are considered to have significant potential for improving treatment outcomes for CRC patients.

The successful development of neoantigen-based therapies hinges upon the identification of neoantigens that exhibit a strong binding affinity to their respective HLA-I molecules and demon-strate high immunogenicity. Initially, DNA sequencing of tumor tissues and paired Peripheral Blood Mononuclear Cells identifies cancer-associated genomic mutations. RNA sequencing then determines the patient's HLA-I allele profile and the gene expression levels of mutated genes. *In silico* tools are then applied to analyze the tumor somatic variant, HLA-I allele, and gene expression data, predicting the binding affinity of neoantigens to the patient's HLA-I alleles and their potential to activate T cell responses (*Hundal et al., 2020*; *Chheda et al., 2018*; *Reynisson et al., 2020*). This standard work-flow has been successful in identifying clinically relevant neoantigens in various malignancies (*Chheda et al., 2018*; *Schumacher and Schreiber, 2015*).

However, despite its achievements, the current approach's impact on patient outcomes remains limited due to the scarcity of mutations in cancer patients that lead to effective immunogenic neoan-tigens. This limitation arises primarily from the fact that the selection of neoantigen candidates by these workflows relies solely on neoantigen-HLA-I binding affinity as the criteria for immunogenicity prediction (*Chen et al., 2021*). While HLA-I binding is indeed a crucial factor for neoantigen presen-tation, it does not fully account for T cell receptor (TCR) recognition and interaction. The recognition of the neoantigen-HLA complex through TCR is of paramount importance for T cell activation and eliciting an immune response (*Szeto et al., 2020*; *Guerder and Flavell, 1995*). Various factors, such as the specific TCR repertoire, TCR clonality, and the structural characteristics of the TCR-peptide-HLA complex, profoundly influence TCR recognition. Unfortunately, these critical aspects are not entirely captured by HLA-I binding prediction alone.

The TCR is a critical component of the adaptive immune system, responsible for recognizing specific antigens presented by antigen-presenting cells (APCs). This membrane-bound heterodimer protein complex is expressed on the surface of T cells and exists in two distinct forms: TCRα/TCRβ for αβ T cells and TCRγ/TCRδ for γδ T cells, both intricately associated with invariant CD3 chain molecules (*Kuhns and Badgandi, 2012*; *Rast, 1997*). TCR diversity arises from the recombination of variable (V), diversity (D), and joining (J) genes at the TCRβ and TCRδ loci, along with VJ recombination at the TCRα and TCRγ loci (*Rosati et al., 2017*), giving rise to a broad array of unique TCRs collectively known as the TCR repertoire. The complementarity-determining region 3 (CDR3), situated at the junc-tion of V, D, and J gene segments (*Wucherpfennig et al., 2010*), plays a pivotal role in antigen recog-nition, with the unique combination of CDR3 sequences contributing significantly to the specificity and diversity of the TCR repertoire. With advancements in technology, TCR sequencing has become a powerful technique used to characterize the diversity and composition of TCR repertoires. Valuable information directly obtained through TCR sequencing, including TCR clonotype diversity, V(D)J gene usage, repertoire size, clonal expansion, and repertoire changes, has provided invaluable insights into immune responses, antigen recognition, and the development of targeted immunotherapies.

Sequencing the TCRs of TILs or lymphocytes found in peripheral blood provides crucial insights into the T-cell repertoire and their responses against neoantigens associated with tumors (*Porciello et al., 2022*; *Lu et al., 2021b*; *Mazzotti et al., 2022*). This information holds paramount importance

in identifying TCRs that specifically target these neoantigens. Additionally, it proves to be invaluable in assessing the immunogenic potential of predicted neoantigens. Unfortunately, the current bioinformatic workflows used in neoantigen selection and prioritization do not incorporate TCR sequencing data. Therefore, we speculate that integrating TCR sequencing data into the assessment of immunogenicity of predicted neoantigens holds promise for unveiling effective neoantigens which can induce immune response, consequently, advancing the development of personalized immunotherapies for cancer.

To achieve this goal, we first performed TCR sequencing on frozen samples collected from CRC patients to profile the TCRβ repertoire of tumor-filtrating T cells. We next exploited the peptide-HLA (pHLA) and pHLA-TCR binding affinity of peptides with known immunogenicity status from six public databases including 10 X Genomics (*10x Genomics, 2024a*; *10x Genomics, 2024b*; *10x Genomics, 2024c*; *10x Genomics, 2024d*), McPAS (*Tickotsky et al., 2017*), PRIME (*Schmidt et al., 2021*), VDJdb (*Shugay et al., 2018*), IEDB (*Vita et al., 2019*), and TBAdb (*Zhang et al., 2020*) to develop a predictive algorithm. Subsequently, we employed this algorithm to predict and rank neoantigens by using the TCRβ, HLA types, and mutation profiles identified from patients' frozen tumor tissues. Finally, we

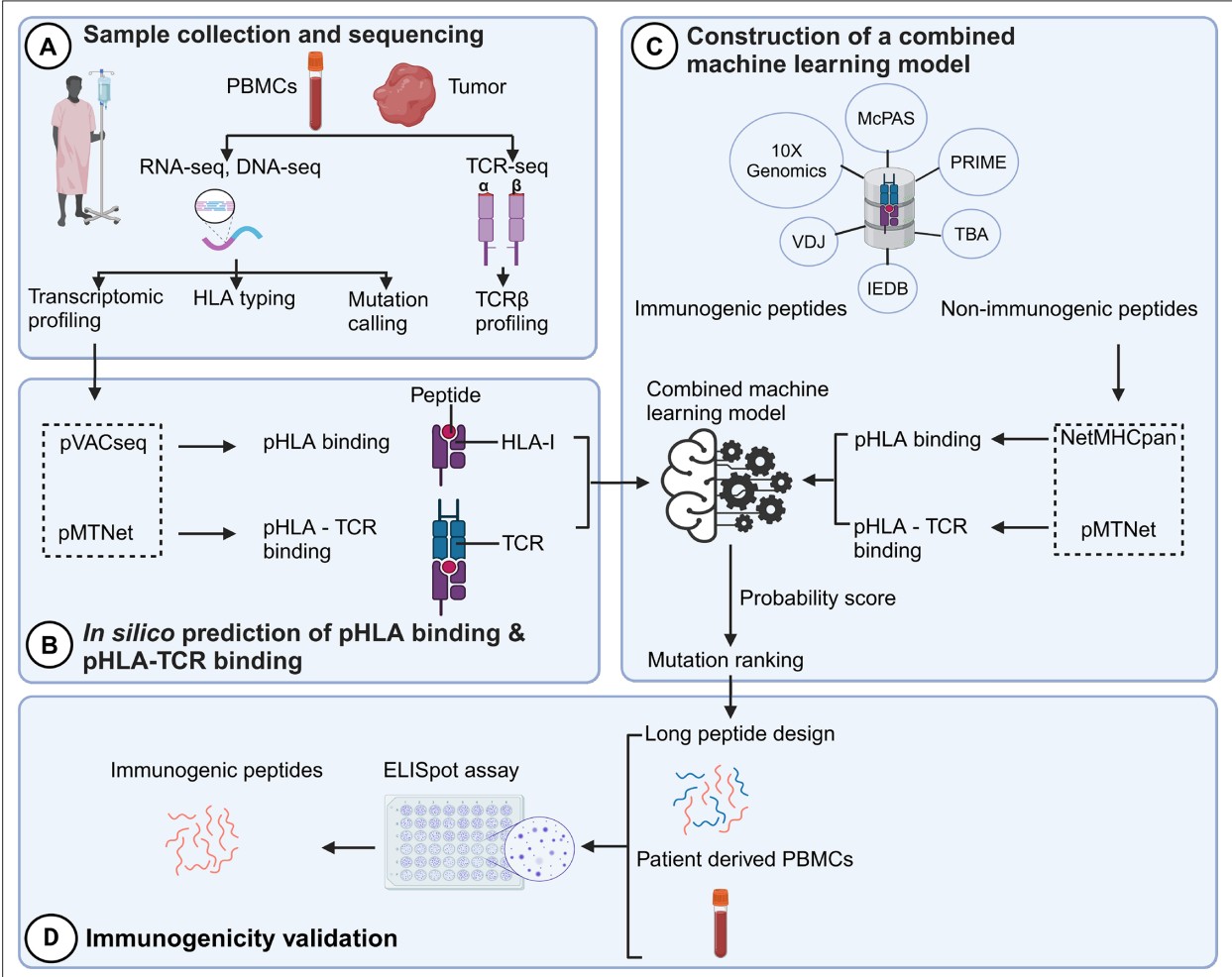

**Figure 1.** A novel workflow based on machine learning that integrates T cell receptor β (TCRβ) sequencing data for the identification and ranking of colorectal cancer (CRC) neoantigens. (**A**) Tumor biopsies and peripheral blood from CRC patients were subjected to targeted DNA-seq, RNA-seq, and T cell receptor (TCR)-seq. (**B**) The prediction of peptide-human leukocyte antigen (HLA) binding and peptide-HLA-TCR binding by indicated tools using the DNA-seq, RNA-seq, and TCR-seq data was performed. (**C**) Machine learning models were subsequently constructed based on the analysis of the peptide-HLA binding and peptide-HLA-TCR binding features to distinguish immunogenic antigens from non-immunogenic peptides. (**D**) The immunogenicity of predicted neoantigen candidates prioritized by the model was validated by enzyme-linked immunospot (ELISpot) to evaluate the effectiveness of this approach.

experimentally verified the efficacy of our model by measuring the immunogenicity of the ranked peptides by an ELISpot assay.

## Results

### The workflow of identifying neoantigens by combining both HLA and TCR binding characteristics

The identification of neoantigens has traditionally heavily relied on pHLA binding prediction while often neglecting the significance of pHLA-TCR interactions (*Lu et al., 2021a*). In this study, we introduce a novel workflow that integrates both pHLA and pHLA-TCR interactions to enhance the precision of neoantigen identification and prioritization (*Figure 1*). In the first step (*Figure 1A*), we conducted RNA and DNA sequencing on matched tumor tissues and peripheral blood mononuclear cells (PBMCs) collected from 28 CRC patients to detect cancer-associated nonsynonymous mutations and determine HLA types, as detailed in our previous work (*Nguyen et al., 2023*). Additionally, T-cell receptor sequencing (TCR-Seq) was performed to profile the TCRβ repertoire of TILs. In the second step (*Figure 1B*), we used the pVACseq and pMTNet tools to predict the binding affinities of both pHLA and pHLA-TCR interactions. To exploit the information of both pHLA and pHLA-TCR binding for selecting and ranking neoantigen candidates, we constructed a machine-learning model by using immunogenic and non-immunogenic peptide information sourced from six publicly available databases (*Figure 1C*). Finally, we designed long peptide (LP) candidates encompassing the selected neoantigens and experimentally assessed their immunogenicity using PBMCs obtained from the same patients (*Figure 1D*).

### Heterogenous tumor infiltrating TCRβ profiles in colorectal cancer patients

Autologous TILs have exhibited varying reactivity levels to neoantigens, suggesting the potential of TIL-based recognition to improve the identification of immunogenic neoantigens (*Chen et al., 2019*). Furthermore, characterizing TCRs can complement efforts to predict immunogenicity. To explore this, we initiated our study by characterizing the TIL repertoire in a cohort of 28 colorectal cancer patients. This characterization involved sequencing the complementarity-determining region 3 (CDR3) of T-cell receptor beta (TCRβ), renowned for its remarkable diversity within the TCR gene. Our sequencing analysis yielded an average of 2,992,949 productive TCR reads per sample, with a range between 256,035 and 10,888,726 (*Supplementary file 1a*), following correction for duplications, sequencing errors, and exclusive use of uniquely barcoded reads mapped to TCRβ-CD3 sequences from the ImMunoGeneTics (IMGT) databases. Across the 28 patients, we observed variable numbers of TCRβ-CDR3 clonotypes, ranging from 433 to 27,749 (*Figure 2A*, *Supplementary file 2A*, 28 patients were arranged in ascending order), indicating the intra-tumor heterogeneity of TCR clonotypes. Of these clonotypes, 59.5% exhibited a single uniquely barcoded read mapped to the TCRβ-CDR3 sequences (depicted in yellow in *Figure 2A*), while 40.5% had at least two TCRβ-CDR3 reads confidently identified (*Figure 2A*). As observed previously (*Hey et al., 2023*), the length distribution of CDR3 was approximately normally distributed (median 14, range 4–43, *Figure 2—figure supplement 1* and *Supplementary file 1b*). Subsequently, we explored potential associations between TCR diversity, as quantified by the Shannon index, and patient characteristics. In accordance with prior investigations, we identified an inverse relationship between the number of TCR clones and the Shannon index (*Figure 2B*). Based on our current sequencing depth, we have observed that many of our samples (14 out of 28) have reached sufficient saturation (*Figure 2C*, *Figure 2—figure supplement 3*) as their diversity of clonotypes was saturated. Additionally, the TCR clones selected in our studies are unique molecular identifier (UMI)-collapsed reads, with each representing at least three raw reads sharing the same UMI. Consequently, despite some samples having low sequencing depth for TCRβ sequencing, likely due to the diversity of TIL infiltration between patients, our TCRβ profiling analysis is robust and reliable. However, we did not detect any significant correlations between TCR clonality or diversity and clinical variables, including microsatellite status, tumor staging, patient gender, and tumor location (*Figure 2—figure supplement 2*). In contrast, MSI-H tumors, which are known to be rich in neoantigens (*Roudko et al., 2020*; *Maleki Vareki, 2018*), displayed a significantly lower number of

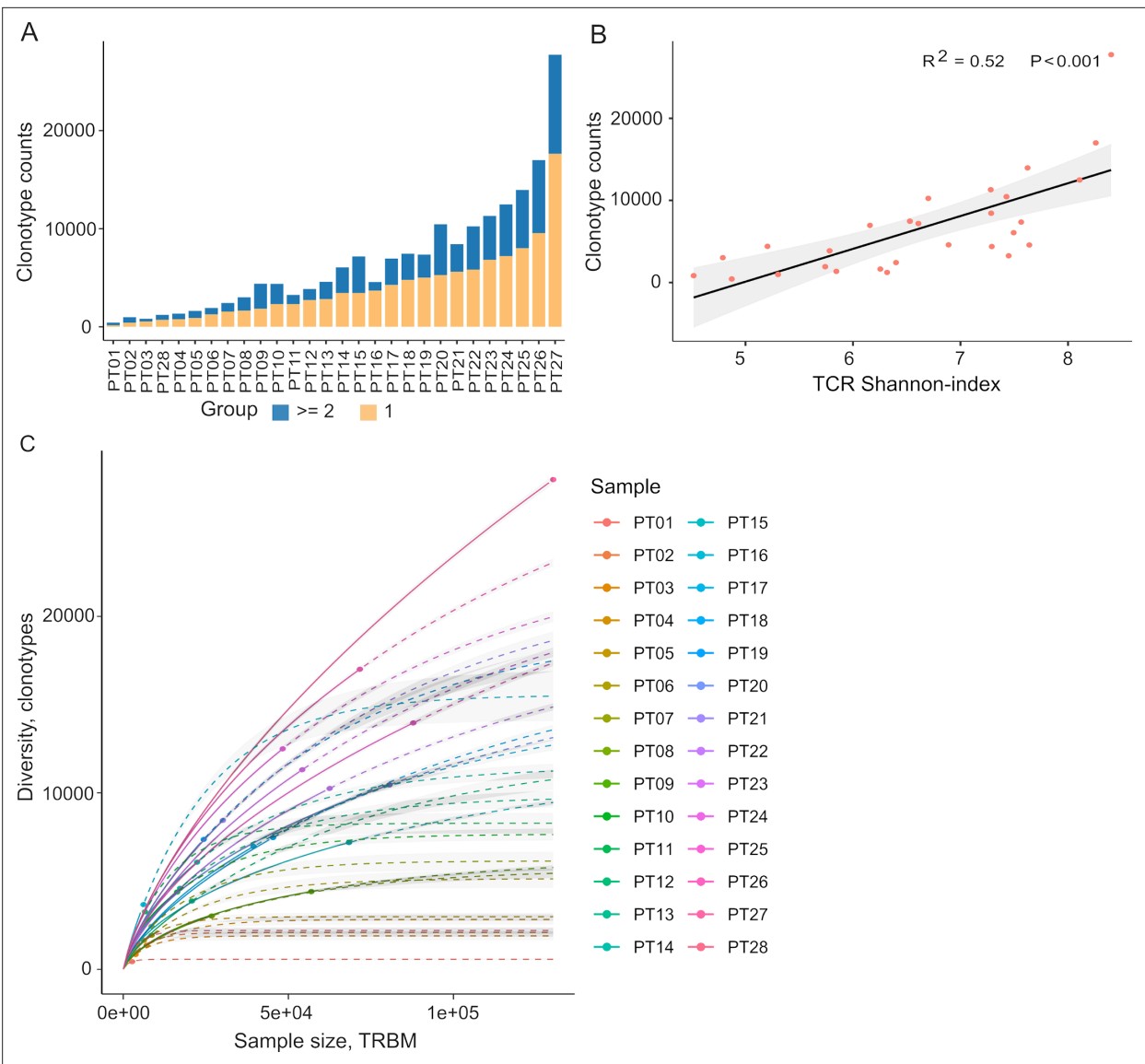

**Figure 2.** Tumor-infiltrating T cell receptor β (TCRβ) profiles in 28 colorectal cancer patients. (**A**) A bar plot depicting the distribution of T cell receptor (TCR) clonotypes among 28 colorectal cancer (CRC) patients, categorized into two groups: those with a unique read count and those with read counts greater than or equal to 2 for each TCR clonotype. (**B**) The scatter plot illustrates the relationship between the Shannon-index and the number of TCR clones. (**C**) The rarefaction plot shows the variable between sample size and diversity among 28 CRC samples.

The online version of this article includes the following figure supplement(s) for figure 2:

**Figure supplement 1.** Quality control metrics for tumor-infiltrating lymphocyte (TIL) T cell receptor β (TCRβ) analysis.

**Figure supplement 2.** Association between tumor-infiltrating lymphocyte (TIL) T cell receptor β (TCRβ) profiles and patients' characteristics.

**Figure supplement 3.** Rarefaction between microsatellite instability (MSI) and microsatellite stability (MSS) samples.

TCRβ clonotypes compared to MSS tumors (2181 vs 5330, p=0.057, *Figure 2—figure supplement 2a and 3*), aligning with previous research (*Laghi et al., 2020*).

After filtering out TCR clones with read counts below 1, we obtained a total of 74,590 TCR clones. Subsequently, we conducted a comprehensive assessment of TCRβ repertoire similarity by calculating the proportion of overlapping TCRβ-CDR3 clones across the 28 patients. The preeminent majority, constituting 95.1% of all identified TCRβ-CDR3 clones, comprised unique clonotypes found in one patient, while the remaining 4.9% were recurrently observed in at least two patients (*Figure 2—figure supplement 2b*). This observation underscores the substantial inter-patient heterogeneity in TCR profiles. The TCRβ repertoire is generated by the random rearrangement of variable (V), diversity

(D), and joining (J) segments. We conducted a search specifically targeting the D segment by scoring the similarities of sub-sequences within the junction sequences against the reference sequences of D segments. Consistent with a previous study, we were unable to unambiguously assign the D segments to any defined D region usage due to the truncation of this region (*Yassai et al., 2009*). In contrast to the diversified D segments, V and J sequments displayed high recurrent rates across 28 patients (*Figure 2—figure supplement 2b*). As anticipated, we identified 59 distinct V segments (*Figure 2—figure supplement 2c*) and 13 distinct J segments (*Figure 2—figure supplement 2d*), collectively sharing 185,627 clones across the 28 tumor tissue samples. This underscores the conservation of these segments (*Figure 2—figure supplement 2c and d*). Conversely, we observed a varied combination of V and J segments, which significantly contributes to the heterogeneity of the TIL TCRβ repertoire (*Figure 2—figure supplement 2e*). Collectively, our data elucidate the presence of both intra-tumor and inter-patient heterogeneity in the TCRβ repertoires of CRC patients, likely stemming from the stochastic utilization of V and J segments in response to neoantigens.

## pHLA and pHLA-TCR interactions are two complement determinants of neoantigen immunogenicity

While *in silico* tools predicting HLA-peptide binding affinity have traditionally played a pivotal role in determining neoantigen immunogenicity (*Vitiello and Zanetti, 2017*), the evaluation of TCR-peptide binding for screening immunogenic neoantigens remains understudied. In light of this, our study aimed to assess the contributions of both pHLA and pHLA-TCR binding affinity in predicting immunogenic neoantigens. To accomplish this, we gathered HLA and TCRβ sequences from established datasets containing immunogenic and non-immunogenic pHLA-TCR complexes (*Supplementary file 1c*). Subsequently, we employed NetMHCpan (*Reynisson et al., 2020*) and pMTNet (*Lu et al., 2021a*) tools to predict pHLA and pHLA-TCR binding, respectively.

For comparative purposes, we generated plots depicting predicted percentile rank values, with lower percentile ranks signifying stronger binding affinity. As anticipated, our analysis revealed a significantly higher prevalence of peptides with robust HLA binding (percentile rank <2%) among immunogenic peptides in contrast to their non-immunogenic counterparts (*Figure 3A & B*, p<0.00001). Similarly, immunogenic peptides exhibited a greater proportion of peptides with percentile ranks indicating pHLA-TCR binding of <2% compared to non-immunogenic peptides (*Figure 3C & D*, p=0.086). As recommended by NetMHCpan and pMTNet, we considered peptide candidates with predicted percentile ranks below 2% as binders. Utilizing this predefined threshold, both pHLA and pHLA-TCR binding affinity exhibited comparable positive predictive values (PPV), at 68.5% (*Figure 3B*) and 64.3% (*Figure 3D*), respectively. This substantiates the significance of both pHLA and pHLA-TCR binding in determining the immunogenicity of peptides. When we simultaneously applied the cutoff values for both HLA-peptide and TCR-peptide percentile ranks (Q1 group, *Figure 3E*), the PPV increased to 76.9% from 68.5% and 64.3% for individual binding features (*Figure 3F*). This underscores the rationale for combining these two criteria to enhance the accuracy of neoantigen prediction. Notably, pHLA-TCR binding displayed a remarkably lower sensitivity but higher specificity in comparison to pHLA binding (*Figure 3G*), implying their potential as a complementary criteria for the selection of immunogenic peptides.

## Combination of peptide-HLA and peptide-HLA-TCR interactions improves neoantigen prediction

The combination of peptide-HLA and peptide-HLA-TCR binding ranking with a fixed cutoff values of 2% for each feature resulted in high specificity but low sensitivity (*Figure 3G*). To optimize the precision of immugenic neoantigens, we examined three distinct machine learning classifiers, namely Random Forest (RF), Logistic Regression (LR), and Extreme Gradient Boosting (XGB) classifiers. We first partitioned the pHLA binding and pHLA-TCR binding ranks for immunogenic and non-immunogenic peptides from publicly available databases into separate discovery (70%) and validation (30%) datasets. These subsets are mutually exclusive and do not overlap (*Figure 4—figure supplement 1*).

These datasets were used to develop and validate machine learning algorithms, as illustrated in *Figure 4A*. We next assessed the performance of the three examined algorithms, employing a 10-fold cross-validation strategy. Among these algorithms, XGB demonstrated the highest performance, achieving an area under the receiver operating characteristic curve (AUC) of 0.82 in the training

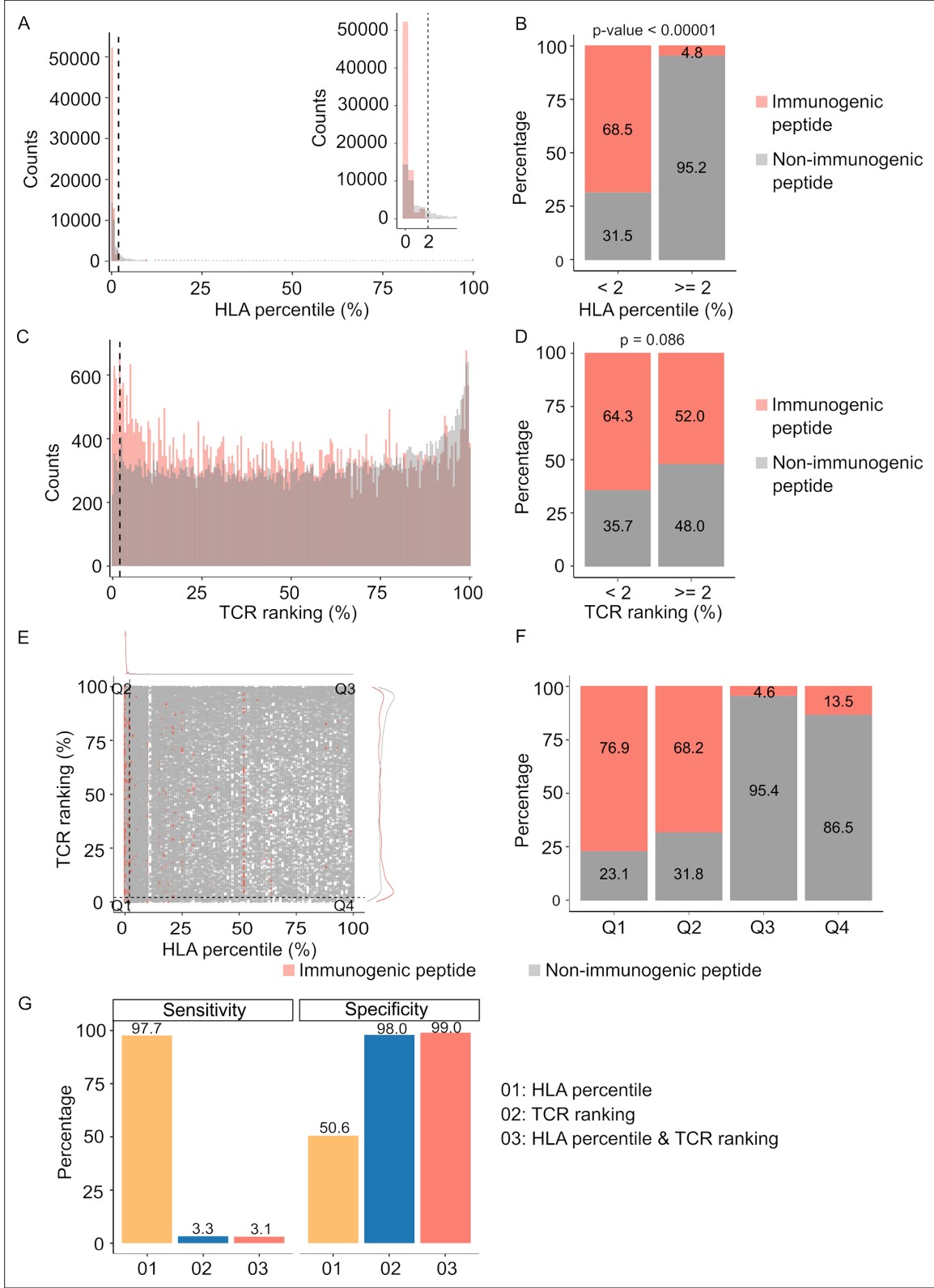

**Figure 3.** Peptide-T cell receptor (TCR) and peptide-human leukocyte antigen (HLA) interactions are two complementary determinants of neoantigen immunogenicity. (**A**) The histogram displays the HLA percentile distribution of immunogenic antigens (red bar) and non-immunogenic peptides (gray bar). (**B**) The percentage of immunogenic antigens (red bar) and non-immunogenic peptides (gray bar) is compared between two groups based on HLA percentile:<2% and ≥ 2% (Chi-square test, p<0.00001). (**C**) The histogram displays the TCR ranking distribution of immunogenic antigens (red

*Figure 3 continued on next page*

*Figure 3 continued*

bar) and non-immunogenic peptides (gray bar). (**D**) The percentage of immunogenic antigens (red bar) and non-immunogenic peptides (gray bar) is compared between two groups based on TCR ranking:<2% and ≥ 2% (Chi-square test, p=0.086). (**E**) The scatter plot illustrates the relationship between the HLA percentile distribution and TCR ranking of immunogenic antigens (red bar) and non-immunogenic peptides (gray bar). (**F**) The percentage of immunogenic antigens (red bar) and non-immunogenic peptides (gray bar) is analyzed in four distinct groups based on cutoffs of HLA percentile and TCR ranking. (**G**) The bar plot illustrates the sensitivity and specificity of three neoantigen prioritization approaches: based on neoantigen-HLA binding affinity alone (yellow bar), neoantigen-TCR binding ranking alone (blue bar), and the combined method using both features (red bar).

dataset and 0.84 in the validation dataset (*Figure 4—figure supplement 2*). Consequently, the model combining pHLA and pHLA-TCR binding, referred to as the 'combined model' was chosen for further validation.

Our findings revealed that the combined model outperformed methods relying on pHLA-TCR or pHLA binding features separately. The combined model yielded sensitivity AUC values of 0.82 (95% CI 0.81–0.84) and 0.84 (95% CI 0.82–0.86) for discovery and validation cohorts, whereas the pHLA-TCR feature alone achieved AUC values of 0.69 (95% CI 0.66–0.71) and 0.74 (95% CI 0.71–0.77). Meanwhile, the pHLA feature resulted in AUC values of 0.76 (95% CI 0.75–0.78) and 0.74 (95% CI 0.72–0.77) (*Figure 4B*). In order to address the elevated false positive rates associated with current prediction tools for neoantigens, we set the specificity at high thresholds of >95% and>99%. We observed that the combined model achieved a greater sensitivity (39.7% vs 5.9% and 19.7% at > 95% specificity; 47.1% vs 3.5% and 18.8% at >99% specificity, *Figure 4C*), negative predictive value (NPV, 87.7% vs 82% and 84.2% at >95% specificity; 88.9% vs 81.5% and 83.9% at >99% specificity, *Figure 4D*), and positive predictive value (PPV, 65.9% vs 22.2% and 47.1% at >95% specificity; 62.3% vs 11.8% and 39.4% at >99% specificity, *Figure 4E*) compared to the single feature methods.

The accurate prioritization of neoantigen candidates with high immunogenicity holds the potential to streamline the validation process, reducing both costs and time expenditures. Consequently, we proceeded to evaluate the combined model's capacity to prioritize neoantigens by computing the ranking coverage score, which considers the accuracy in ranking immunogenic peptides versus non-immunogenic peptides (*Zhou et al., 2019b*). In the validation phase, the combined model exhibited superior rank coverage scores in comparison to the individual feature-based methods. The combined model attained a ranking coverage score of 0.37, while the single-feature methods, pHLA-TCR and pHLA, yielded scores of –0.26 and 0.25, respectively (*Figure 4F*). These findings underscore the notion that the incorporation of pHLA and pHLA-TCR binding criteria can enhance the accuracy of prediction and prioritization of immunogenic neoantigens.

## Experimental validation of the pHLA-TCR and pHLA combined approach in selecting neoantigen candidates in patients with CRC

To experimentally validate the efficacy of our combined approach, we conducted a comparative analysis of the percentage of confirmed immunogenic neoantigen candidates and the ranking coverage scores between the conventional NetMHCpan method, which relies solely on pHLA binding percentiles, and our combined approach (*Figure 5A*). Due to the limited availability of patient blood samples, we were only able to perform validation on eight patients who possessed a sufficient quantity of PBMCs. For each patient, we chose the top three neoantigen candidates predicted by each method. Among the neoantigen candidates, three were found to be common between the two methods, while twenty neoantigen candidates were uniquely identified by either NetMHCpan or the combined model (*Figure 5B*), bringing the total number of candidates to 44. Subsequently, we synthesized 44 LPs covering these 44 selected neoantigen candidates, along with 44 LPs corresponding to the wild-type sequences. The LPs were utilized in an *ex vivo* ELISpot assay to measure the release of interferon-gamma (IFNγ) from the four patients' PBMCs. If the stimulation with a mutant LP resulted in a >twofold increase in IFNγ spots compared to its corresponding wild-type LP, the respective neoantigen candidate was classified as an immunogenic neoantigen.

Out of the 44 selected LPs, we confirmed the immunogenicity of three LPs (RNF213_W719S, MAP3K1_L1202F, and TRRAP_T148TEL) identified by the NetMHCpan method and seven LPs (NCOR2_R1963H, TRRAP_F1568S, DICER1_P1592L, KMT2C_K3848T, BRAF_EV275-276-, SMAD4_G230R, and PTPN13_D184A) selected by the combined method (*Figure 5C and D*). Notably, six patients exhibited at least one immunogenic peptide identified by the combined method, whereas

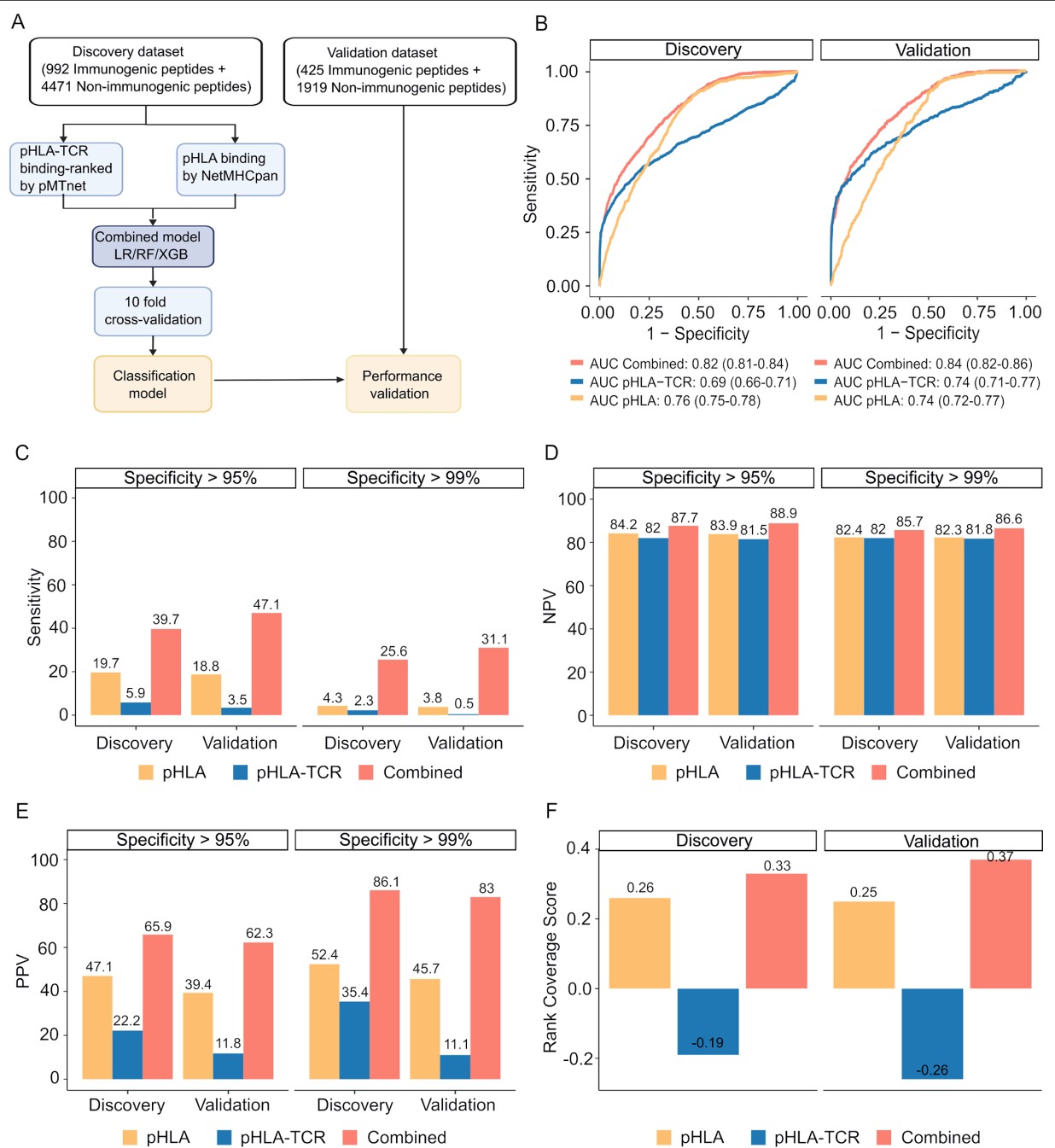

**Figure 4.** The combined model demonstrates improved sensitivity and specificity for neoantigen prioritization. (**A**) The workflow for constructing the model. (**B**) The receiver operating characteristic (ROC) curves demonstrate the performance of both the combined model and individual models in both the discovery and validation cohorts. The bar graphs illustrate the sensitivity (**C**), negative predictive value (NPV) (**D**), and positive predictive value (PPV) (**E**) at specificity levels of at least 95% or 99% for both the combined and individual models in both the discovery and validation cohorts. (**F**) Ranking coverage scores for the specified models in either the discovery or validation cohorts.

The online version of this article includes the following figure supplement(s) for figure 4:

**Figure supplement 1.** Dataset construction workflow.

**Figure supplement 2.** The performance of three machine learning models with three different algorithms is evaluated using receiver operating characteristic (ROC) curves.

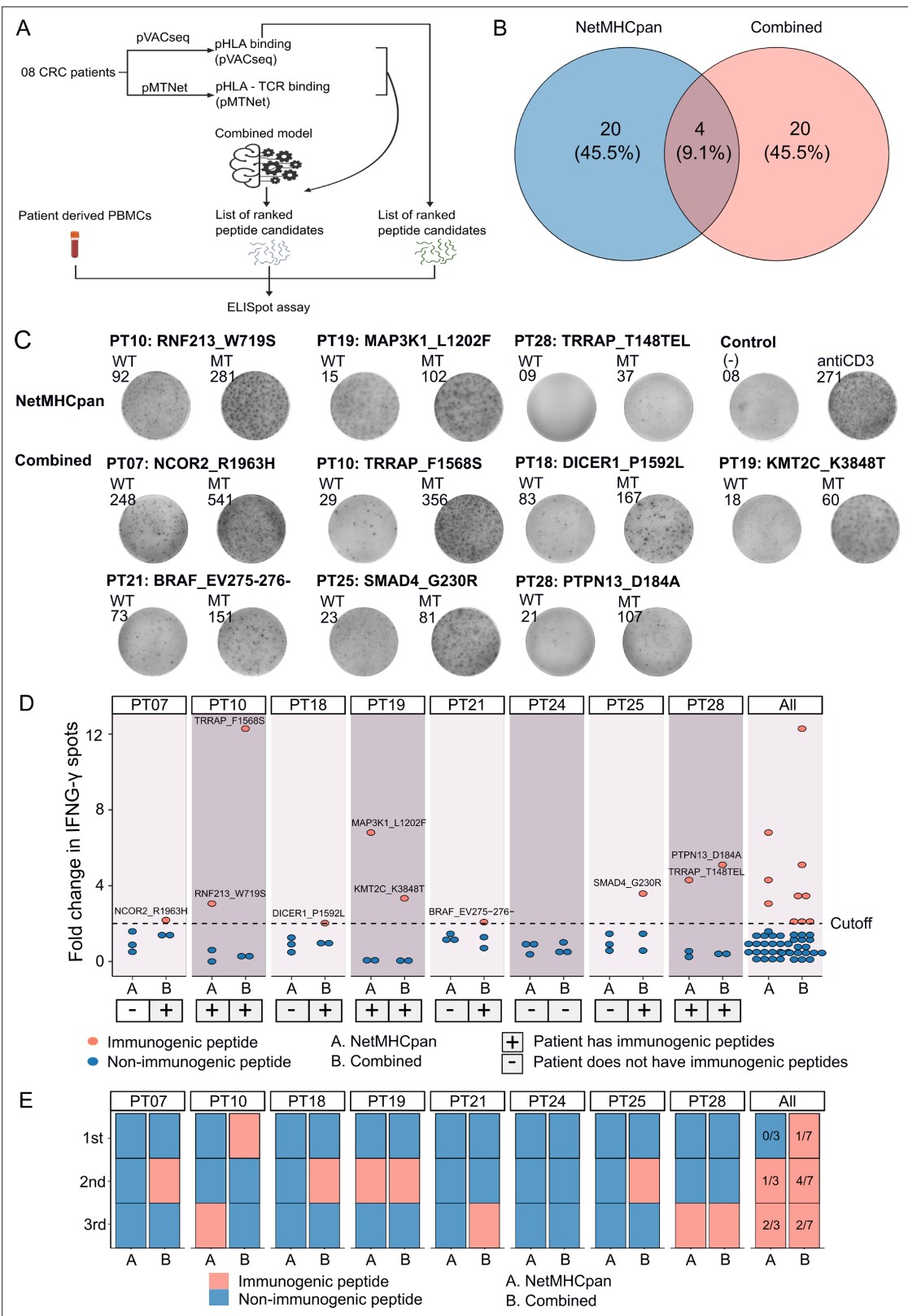

**Figure 5.** Validation of neoantigens identified in silico from the novel workflow through enzyme-linked immunospot (ELISpot) assays conducted on four colorectal cancer (CRC) patients. (**A**) A schematic diagram illustrates the procedural steps of neoantigen prioritization and the ELISpot assay. (**B**) The count of neoantigens identified from each pipeline. (**C**) The fold change in IFN-γ spots, relative to the wild-type peptides, is shown for 21 long peptides. Note: Only the mutants that result in a positive value in ELISpot are depicted, along with their corresponding amino acid changes and their associated

*Figure 5 continued on next page*

*Figure 5 continued*

rankings. (**D**) ELISpot assays on six long peptides resulting in at least a twofold change in IFN-γ spots. (**E**) The bar graphs display the ranking of validated long peptides identified from the NetMHCpan tool (blue bar) or the combined method (red bar) for individual patients and all patients.

The online version of this article includes the following figure supplement(s) for figure 5:

**Figure supplement 1.** The rank coverage score of the combined model compared to NetMHCpan.

none of the LPs in five patients, PT07, PT18, PT21, PT24, and PT25, chosen via the NetMHCpan method were validated as immunogenic (*Figure 5C and D*). To further assess the ranking accuracy of these two methods, we calculated the ranking coverage scores. In agreement with our in silico analyses, we observed higher rank coverage scores for the combined method in five out of the eight patients, resulting in an overall score of 0.04 (*Figure 5E*, *Figure 5—figure supplement 1*). In contrast, the NetMHCpan method exhibited a lower rank coverage score of –0.37 (*Figure 5E*, *Figure 5—figure supplement 1*). To further evaluate our model, we gathered additional public data and assessed its effectiveness in comparison to other models. We utilized immunogenic peptides from databases such as NEPdb (*Xia et al., 2021*), NeoPeptide (*Zhou et al., 2019a*), dbPepneo (*Tan et al., 2020*), Tantigen (*Zhang et al., 2021*), and TSNAdb (*Wu et al., 2018*), ensuring there was no overlap with the datasets used for training and validation. For non-immunogenic peptides, we used data from 10 X Genomics Chromium Single Cell Immune Profiling (*10x Genomics, 2024a*; *10x Genomics, 2024b*; *10x Genomics, 2024c*; *10x Genomics, 2024d*). The findings indicate that the combined model from pMTNet and NetMHCpan outperforms the NetTCR tool (*Montemurro et al., 2021*; *Supplementary file 1d*). These outcomes conclusively demonstrate the ability of the combined approach to enhance the prediction and ranking of immunogenic neoantigens in cancer patients.

## Discussion

The selection of neoantigens plays a pivotal role in enhancing the efficacy of personalized treatments in cancer immunotherapy. Historically, neoantigen selection has predominantly hinged upon the prediction of pHLA binding. However, the limitations of this approach have become increasingly evident (*Borden et al., 2022*), as it often neglects the dynamic interplay between tumor cells and the immune system. The findings of our study shed new light on this critical aspect of neoantigen selection. Our research highlights the potential of integrating pHLA binding prediction with the assessment of pHLA-tumor-infiltrating Lymphocyte T Cell receptor (TIL TCR) binding (*Figure 1*). By encompassing the interaction between neoantigens and the tumor microenvironment, we have demonstrated a substantial enhancement in the accuracy of neoantigen selection and prioritization. Our findings underscore the significance of comprehensive neoantigen assessment in harnessing the full potential of immunotherapeutic strategies in the fight against cancer.

Our investigation into the TIL TCRβ repertoire across 28 CRC patients has unveiled a complex tapestry of intra-tumor and inter-patient heterogeneity (*Figure 2A*, *Figure 2—figure supplement 1*). One underlying cause of this diversity is the random rearrangement of variable (V) and joining (J) segments, which contributes to the distinct TCRβ sequences observed (*Figure 2—figure supplement 1c, d & e*). This remarkable diversity in the TCRβ profile, as elucidated in our study, presents a potential link to the heterogeneity of cancer mutations and neoantigens within CRC. The intricate relationship between the TCRβ repertoire and the genetic alterations within the tumor microenvironment underscores the need for a more comprehensive understanding of this dynamic interplay to develop more precise immunotherapeutic strategies (*Joshi et al., 2019*). Furthermore, our research revealed a compelling observation regarding patients with MSI-H, who are known to exhibit high mutation and neoantigen burdens (*Motta et al., 2021*; *Xie et al., 2023*). Intriguingly, MSI-H patients with strong immune reactivity to neoantigens was shown to display a lower number of TCR clonotypes and reduced diversity, as characterized by the Shannon index, compared to CRC patients with microsatellite stable (MSS) (*Figure 2—figure supplement 2*). This stark contrast in TCRβ diversity hints at the potential enrichment of neoantigen-reactive TCR clonotypes in MSI-H patients. These findings provide valuable insights into the immune responses to different subtypes of CRC and may guide the development of more tailored immunotherapies for patients with differing mutational landscapes.

By utilizing proven immunogenic and non-immunogenic peptides from public databases, we have demonstrated that predicting the strength of pHLA-TCR binding can be a crucial criterion for

selecting immunogenic candidates (*Figure 3*). For the prediction of pHLA-TCR binding, we employed the well-established tool pMTnet (*Lu et al., 2021a*). This choice was driven by its proven efficacy, based on its consistently high performance across different validated datasets (*Lu et al., 2021a*). Although it exhibited lower sensitivity and PPV when compared to pHLA binding alone, pHLA-TCR binding strength exhibited notably superior specificity (*Figure 3G*). This implies that incorporating pHLA-TCR binding strength as a selection criterion would result in a reduced false positive rate, a crucial factor in the context of neoantigen selection. What makes our study particularly compelling is the convergence of these observations. The integration of both criteria, specifically, pHLA binding and pHLA-TCR binding strength, emerged as a strategy that not only enhanced the PPV but also fortified the rationale for combining these two features in the neoantigen selection process.

The integration of both pHLA binding and pHLA-TCR binding strength features in our approach exhibited superior performance in neoantigen selection and prioritization when compared to the single-feature method (*Figure 4*). This observation aligns with the findings of previous studies, which consistently indicate that combining multiple criteria enhances the accuracy and efficacy of neoantigen identification (*Zhou et al., 2019b*; *Müller et al., 2023*). Furthermore, our experimental validation confirmed the robustness of our approach by consistently demonstrating performance consistent with our analysis of publicly available data (*Figure 5*, *Supplementary file 1d*). When examining the percentile ranks of positive LPs predicted by netMHCpan from those predicted by our combined model, the results further underscored the strength of our approach (*Figure 5E*). This alignment between experimental validation and computational analysis enhances the reliability and applicability of our neoantigen selection method. Interestingly, the four identified neoantigen candidates with confirmed immunogenicity by the combined approach have not been previously reported in public neoantigen-databases, indicating that they could serve as novel targets for neoantigen based therapies.

However, several limitations must be acknowledged. Firstly, the relatively small sample size employed for validation raises the potential for ranking score bias. Additionally, while both TCRα and TCRβ regions play essential roles in engaging peptide-bound HLA complexes, our study focused solely on TCRβ sequences to predict pHLA-TCR binding strength. Future investigations should include TCRα sequences to provide a more comprehensive analysis. Although the PBMC and TILs were previously shown to be congruent in neoantigen reactivity (*Malekzadeh et al., 2020*), it is important to recognize that differences in the contribution of TIL and PBMC TCR repertoires in neoantigen selection may introduce variability in the selection and validation of neoantigen candidates, and future studies are warranted to address the consistency in neoantigen reactivity between these two sources of T cells. Moreover, to improve the accuracy and effectiveness of the machine learning model in predicting and ranking neoantigens, we have developed an in-house tool called EpiTCR. This tool will utilize immunogenic assays, such as ELISpot and single-cell sequencing, for validation.

In summary, our study delves into the diversity and variation within the TCRβ repertoire of TILs in the tumor tissues of colorectal cancer patients. Through our research, we have introduced a novel approach to the identification and prioritization of neoantigen candidates. This approach combines the assessment of pHLA binding and peptide-human leukocyte antigen-T cell receptor (pHLA-TCR) binding. Our findings underscore the significance of considering pHLA-TCR binding interactions as part of the process for selecting neoantigen candidates. This is a crucial step in the development of personalized immunotherapy strategies. By combining these two factors, we can more accurately identify the neoantigens that hold promise for effective immunotherapy, ultimately improving the prospects for tailored and effective cancer treatment.

## Materials and methods
### Tumor biopsy and peripheral blood collection
Twenty-eight patients diagnosed with CRC were enrolled in this study at the University Medical Center in Ho Chi Minh City between June 2022 and May 2023. CRC confirmation was based on abnormal colonoscopies and histopathological analysis. The stages of CRC were determined according to guidelines provided by the American Joint Committee on Cancer and the International Union for Cancer Control (the eighth version) (*Amin et al., 2017*; *Tong et al., 2018*). Prior to participation, all patients provided written informed consent for tumor and whole blood sample collection. Relevant clinical data, including demographics, cancer stages, and pathology information, were extracted

from the medical records of the University Medical Center. Detailed information regarding the clinical factors of the patients can be found in *Supplementary file 1e*. This study was approved by the Ethics Committee of the University of Medicine and Pharmacy in Ho Chi Minh City, Vietnam. For eight patients, ten mL of peripheral blood was collected before surgery and stored in Heparin tubes prior to isolation of PBMCs.

## Targeted DNA, RNA, and TCRβ sequencing

The DNA/RNA samples were isolated using either the AllPrep DNA/RNA Mini Kit or the AllPrep DNA/RNA/miRNA Universal Kit (Qiagen, Germany) as per the manufacturer's protocol. In addition, matched genomic DNA from the white blood cells (WBC) of individuals was also extracted from the buffy coat using the GeneJET Whole Blood Genomic DNA Purification Mini kit (Thermo Fisher, MA, USA), following the manufacturer's instructions. Genomic DNA samples from the patients's paired tumor tissues and WBCs were used to prepare DNA libraries for DNA sequencing with the ThruPLEX Tag-seq Kit (Takara Bio, USA). The libraries were then pooled and hybridized with pre-designed probes for 95 targeted genes (Integrated DNA Technologies, USA). This gene panel encompasses commonly mutated genes in CRC tumors, as reported in the Catalog of Somatic Mutations in Cancer (COSMIC) database. The DNA libraries were then subjected to massive parallel sequencing on the DNBSEQ-G400 sequencer (MGI, Shenzhen, China) for paired-end reads of 2×100 bp with a sequencing depth of 10 X.

Isolated total RNA was processed using the NEBNext Poly(A) mRNA Magnetic Isolation Module (New England Biolabs, MA, USA) to isolate intact poly(A)+ RNA following the manufacturer's instructions. RNA libraries were prepared using the NEBNext Ultra Directional RNA Library Prep Kit for Illumina (New England Biolabs). These libraries were subsequently sequenced for paired-end reads of 2×100 bp on an MGI system at a sequencing depth of 50 X.

For TCRβ library construction, mRNA was utilized with the SMARTer Human TCR a/b Profiling Kit v2 (Takara, USA). The resulting libraries were sequenced for paired-end reads of 2×250 bp on an Illumina system at a sequencing depth of 7.5 X.

## Microsatellite instability status

Microsatellite instability (MSI) status was determined in both tumor DNA and corresponding DNA from blood, serving as the control, using the MSI Analysis System, version 1.2 (Promega, US). This analysis identified alterations in five mononucleotide repeat markers: BAT-25, BAT-26, NR-21, NR-24, and MONO-27. The results were classified as MSS in cases where either none or only one marker was unstable. Samples showing alterations in two or more markers were categorized as microsatellite instability-high (MSI-H).

## Variant calling from DNA-seq and RNA-seq data

The approach of integrating RNA-seq into calling variants was performed as described in our previous publication (*Nguyen et al., 2023*). Briefly, we used Dragen (Illumina) (*Catreux et al., 2022*) in tumor-normal mode to detect somatic mutations from DNA-seq data. The default filtering thresholds of Dragen were applied for the detection of single nucleotide polymorphisms (SNPs) and insertions/deletions (indels). SNPs were further filtered using the dbSNP and 1000 Genome datasets. Germline mutations in tumor tissues were identified by comparison with matched WBC DNA samples. Mutations within immunoglobulin and HLA genes were excluded due to alignment difficulties in these highly polymorphic regions, necessitating specialized analysis tools. Additionally, synonymous mutations were removed from downstream analysis. For analysis, we included somatic mutations that exceeded a minimum threshold of ≥2% variant allele frequency (VAF) in DNA extracted from fresh frozen tissues.

Sequencing reads underwent trimming using Trimmomatic (*Bolger et al., 2014*) and were then aligned to the human reference genome using STAR (version 2.6.0 c) (*Dobin, 2013*). Prior to alignment, raw sequencing reads were subjected to quality checks using FastQC version 0.11.9 (*Andrews, 2010*). VarScan 2 (*Koboldt et al., 2012*), which accepts both DNA and RNA-seq data, was employed to identify mutations in paired tumor and WBC samples across 95 cancer-associated genes. This analysis was carried out in tumor-normal mode. Four filtering steps were applied: (i) only calls with a PASS status were utilized, (ii) population SNPs overlapping with a panel of normal samples from the 1000 Genome dataset were excluded, (iii) somatic mutations included for analysis met a minimum threshold

of ≥10x read depth and ≥2% variant allele frequency (VAF) in RNA extracted from FF tissue, and (iv) synonymous mutations and those related to HLA were removed from downstream analysis.

The resulting BAM files were sorted, and indexed using Samtools version 1.10 (*Li et al., 2009*), and had PCR duplicates removed using Picard tools version 2.25.6 (*Picard, 2024*). Somatic variants were manually reviewed using Integrative Genomics Viewer (v2.8.2). The VCF files generated by Dragen (for DNA-seq) and VarScan 2 (for RNA-seq) were subsequently annotated using the Ensembl Variant Effect Predictor (VEP version 105) (*McLaren et al., 2016*) to extract information about the potential effects of variants on the phenotypic outcome.

### CDR3β calling from TCRseq data

To define the CDR3β from TCRseq, we utilized Cogent NGS Immune Profiler Software v1.0 (*Takara Bio, 2024*). Before calling clonotypes, the raw sequencing reads underwent filtering based on the following criteria: (i) allowing only one mismatch while splitting reads by matching read sequences to different receptor chains, (ii) excluding reads shorter than 30 bp and reads ambiguously matched to multiple receptor chains, (iii) excluding reads that failed correction when linker-based correction was enabled, (iv) excluding reads that failed the abundance check during sequencing error correction, and (v) excluding molecular identifier groups (MIGs) with fewer than three unique molecular identifier (UMI) reads. The filtered MIGs were then assembled and aligned to the V(D)J reference to define the TCR clonotype.

### Shannon index and clonality

We utilized two indices to characterize T-cell diversity and expansion: the Shannon entropy index and the clonality index.

Shannon entropy:

$$-\sum_{i=1}^{n} p_i \log_e \left( p_i \right)$$

Clonality:

$$1 - \frac{\sum_{i=1}^{n} p_i \log_e \left( p_i \right)}{\log_e \left( n \right)}$$

These indices consider both the number of T-cell clone types 'n' and the frequency '$p_i$' of each clone. Here, '$p_i$' represents the proportion of the i-th clone within the TCR library containing n clones.

### *In silico* prediction of peptide-HLA binding and peptide-HLA-TCR binding

Class I HLA alleles (HLA-A/B/C) with two-digit resolution were identified from patient tumor RNA-seq data using the OptiType tool (*Supplementary file 1f*; *Szolek et al., 2014*; *Li et al., 2024*). The annotated VCF files were analyzed using pVACseq, a tool in pVACtools (v1.5.9) (*Hundal et al., 2020*; *Hundal et al., 2016*; *Hundal et al., 2019*). We used the default settings, except for disabling the coverage and MAF filters. We used all peptide-HLA-I binding algorithms that were implemented in pVACseq to predict 8–11-mer epitopes binding to HLA-I (A, B, or C) for downstream analysis. Mutants with lower binding affinity than wild-type peptides were prioritized using a two-step ranking process. The minimum binding affinity score was calculated from the distribution of scores among the peptides derived from each mutation, and priority was given to mutations with the lowest binding affinity scores. Moreover, in pVACseq, we collected peptides with binding affinity scores lower than wild-type from multiple tools, calculated minimum binding affinity scores for each unique peptide, and incorporated this data into a combined machine learning model.

We used the peptide-HLA-TCR binding algorithm implemented in pMTNet (*Lu et al., 2021a*) to predict peptide binding to HLA and TCR with default settings. These scores represented the predicted likelihood of peptides being immunogenic. A ranking value for immunogenicity was assigned to each unique peptide by determining the minimum TCR ranking of its immunogenicity score.

## Construction of a combined machine-learning model

To identify immunogenic peptides, we conducted a thorough search across multiple databases, including 10 X Genomics (*10x Genomics, 2024a*; *10x Genomics, 2024b*; *10x Genomics, 2024c*; *10x Genomics, 2024d*), McPAS (*Tickotsky et al., 2017*), VDJdb (*Shugay et al., 2018*), IEDB (*Vita et al., 2019*), and TBAdb (*Zhang et al., 2020*). For the creation of a non-immunogenic pHLA-TCR complex, we assembled a negative dataset using the PRIME tool (*Schmidt et al., 2021*). This dataset was generated by associating each pHLA with 10 randomly generated TCRs sourced from the following databases: 10 X Genomics, McPAS, VDJdb, IEDB, and TBAdb. To objectively train and evaluate the model, we separated the dataset mentioned above into two data subsets: discovery dataset (70%) and validation dataset (30%). These subsets are mutually exclusive and do not overlap (*Figure 4—figure supplement 1*). pHLA-TCR complex in the discovery dataset, whether labeled as immunogenic or non-immunogenic, were used for model training to classify whether a peptide is positive or not. We examined three machine learning algorithms - Logistic Regression (LR), Random Forest (RF), and Extreme Gradient Boosting (XGB) - for each feature type (pHLA binding, pHLA-TCR binding), as well as for combined features. Feature selection was tested using a k-fold cross-validation approach on the discovery dataset with 'k' set to 10-fold. This process splits the discovery dataset into 10 equal-sized folds, iteratively using ninefolds for training and onefold for validation. Model performance was evaluated using the 'roc_auc' (Receiver Operating Characteristic Area Under the Curve) metric, which measures the model's ability to distinguish between positive and negative peptides. The average of these scores provides a robust estimate of the model's performance and generalizability. The model with the highest 'roc_auc' average score, XGB, was chosen for all features. The model cut-off was set based on a threshold specificity of >90% or>95% to achieve high specificity for selecting peptides. This combined model's performance was evaluated on the independent validation dataset of immunogenic and non-immunogenic pHLA-TCR complex to assess its ability to classify positive peptides from negative peptides. In our GitHub repository, we have included links to the GitHub repositories for NetMHCpan and pMTNet (https://github.com/QuynhPham1220/Combined-model copy archived at *Pham, 2024*).

## Ranking coverage score calculation

The approach of compare ranking between two algorithms was performed as described in a previous publication (*Zhou et al., 2019b*). Briefly, the rank coverage score was based on the ranking value calculation given by the formula:

$$\text{Rank Coverage Score} = \frac{\sum_{n \in \text{negative}} \text{rank(n)}}{T \times \text{num(n)}} \times \text{coverage(n)} - \frac{\sum_{n \in \text{positive}} \text{rank(p)}}{T \times \text{num(p)}} \times \text{coverage(p)}$$

$$\text{coverage(k)} = \frac{\max \text{rank(k)}}{T} k \in (n, p)$$

Where, *T* presented the total neoepitope number identified and *p* and *n* presented the positive and negative peptides, respectively, that were experimentally validated *in vitro*.

If positive peptides have smaller rank values than negative peptides, this will result in a high-rank coverage score, indicates the better ranking result.

## Isolation, culture, and stimulation of PBMCs with long peptides

Peripheral blood samples were collected from eight patients prior to surgery using BD Vacutainer Heparin Tubes (BD Biosciences, NJ, USA). PBMCs were isolated through gradient centrifugation using Lymphoprep (STEMCELL Technologies) within 4 hr of blood collection. The PBMCs were then resuspended in a solution of FBS (10%) and DMSO ($7–10 \times 10^6$ cells/mL) for cryopreservation in liquid nitrogen. Frozen PBMCs were thawed in AIM-V media (Gibco, Thermo Scientific, MA, USA) supplemented with 10% FBS (Cytiva, USA) and DNase I (Stemcell Technology, Canada) (1 mg/mL) solution. A total of $10^5$ PBMCs were allowed to rest in a 96-round bottom well-plate containing AIM V media supplemented with 10% FBS, 10 mM HEPES, and 50 mM β-mercaptoethanol overnight before stimulation with synthesized long peptides at a concentration of 5 mM in a humidified incubator at 37 °C with 5% CO2. PBMCs were further stimulated with GM-CSF (2000 IU/mL, Gibco, MT, USA) and IL-4 (1000 IU/mL, Invitrogen, MA, USA) for 24 hr. Following this initial stimulation, LPS (100 ng/mL, Sigma-Aldrich, MA, USA) and IFN-γ (10 ng/mL, Gibco, MT, USA) were added to the PBMCs along with the

peptides for an additional 12 hr. On the following day, IL-7, IL-15, and IL-21 (each at a concentration of 10 ng/mL) (Peprotech, NJ, USA) were added to the PBMC culture. The restimulation process involved exposing the peptides to fresh media containing IL-7, IL-15, and IL-21 every 3 days, for a total of three times. On day 12, PBMCs were restimulated with peptides and cultured in media without cytokines. ELISpot assays were performed on stimulated PBMCs on day 13.

## ELISpot assay on PBMCs stimulated with long peptides

Cultured T cells were transferred to an enzyme-linked immunospot (ELISpot) plate (Mabtech, Sweden) and incubated for 20 hr at 37 °C. PBMCs cultured with DMSO were used as a negative control group, while PBMCs stimulated with anti-CD3 were used as a positive control group. The ELISpot assay was performed on treated PBMCs using the ELISpot Pro: Human IFN-g (ALP) kit (Mabtech, Sweden), following the manufacturer's protocol. Developed spots on the ELISpot plate were then counted using an ELISpot reader (Mabtech, Sweden). Reactivity was determined by measuring the fold increase in the number of spots in PBMCs treated with mutant peptides compared to those treated with wild-type peptides. A fold change of two was selected as the cutoff for positivity, indicating a significant increase in reactivity (*Moodie et al., 2010*).

## Statistical analysis

In this study, t-tests were used to compare the TCR clones, clonality, and Shannon index among two groups of four categories (Microsatellite status, stage, gender, and tumor location). Chi-square test was used to compare the proportions of immununogenic and non-immunogenic peptides. All statistical analyses were performed using R (version 4.3.0) with common data analysis packages, including ggplot2 and pROC. The 95% confidence interval (95% CI) was presented in brackets next to a value as appropriate.

## Acknowledgements

The authors thank all participants who agreed to take part in this study. We thank Dr. Kien Nguyen for proofreading our manuscript.

## Additional information

### Funding

| Funder | Grant reference number | Author |
| --- | --- | --- |
| NextCalibur Therapeutic | NC01 | Le Son Tran<br>Minh-Duy Phan |

The funders had no role in study design, data collection and interpretation, or the decision to submit the work for publication.

### Author contributions

Thi Mong Quynh Pham, Data curation, Formal analysis, Methodology, Writing - original draft, Writing - review and editing; Thanh Nhan Nguyen, Bui Que Tran Nguyen, Thi Phuong Diem Tran, Nguyen My Diem Pham, Hoang Thien Phuc Nguyen, Thi Kim Cuong Ho, Dinh Viet Linh Nguyen, Thi Tuong Vy Nguyen, Data curation, Formal analysis, Methodology; Huu Thinh Nguyen, Conceptualization, Patient Consultation; Duc Huy Tran, Patients recruitment and tissue histological analysis; Thanh Sang Tran, Patients recruitment and tissue histological analysis; Truong Vinh Ngoc Pham, Patients recruitment and tissue histological analysis; Minh Triet Le, Patients recruitment and tissue histological analysis; Minh-Duy Phan, Le Son Tran, Conceptualization, Writing - review and editing; Hoa Giang, Conceptualization, Methodology; Hoai-Nghia Nguyen, Conceptualization, Supervision

### Author ORCIDs

Thi Mong Quynh Pham (ORCID) http://orcid.org/0000-0002-6977-6064
Huu Thinh Nguyen (ORCID) https://orcid.org/0000-0002-0981-7274
Thi Tuong Vy Nguyen (ORCID) https://orcid.org/0000-0003-3436-3662

Le Son Tran https://orcid.org/0000-0002-5382-3903

Reviewer #2 (Public Review): https://doi.org/10.7554/eLife.94658.3.sa1
Author response https://doi.org/10.7554/eLife.94658.3.sa2

## Additional files

### Supplementary files

• Supplementary file 1. Quality metrics of TCRb sequencing.

• Supplementary file 2. Percentage of unique and expanded T cell receptor (TCR) clones from 28 colorectal cancer (CRC) patients.

• Supplementary file 3. List of immunogenic peptides and non-immunogenic peptides from public databases.

• Supplementary file 4. Performance of combined model in an independent dataset.

• Supplementary file 5. Clinical characteristics of 28 colorectal cancer (CRC) patients.

• Supplementary file 6. Comparison of human leukocyte antigen (HLA) calling accuracy between Acras-HLA and Optitype.

### Data availability

The Vietnamese government has enacted a new law on the Protection of Personal Data (Decree No. 13/2023/ND-CP), which came into effect on July 1, 2023. This law includes specific requirements for the storage and sharing of personal data, including genetic information. Consequently, researchers interested in using our genetic data must apply for approval before accessing it.To request access to sequencing data (FastQC files) from individuals, please contact the senior author, Dr. Le Son Tran, Medical Genetics Institute (MGI), at 186-188 Nguyen Duy Duong Street, District 10, Ho Chi Minh City, Vietnam, or via email at leson1808@gmail.com. Researchers must submit a data-sharing agreement, agreeing to the terms and conditions outlined therein. The Director of the Medical Genetics Institute will review and approve all data-sharing agreements. There are no restrictions on access to the raw data; both commercial and academic researchers may use the data for their respective studies. The code used in this study is publicly available at https://github.com/QuynhPham1220/Combined-model, copy archived at *Pham, 2024*. Other code and software used in this publication are mentioned in the methodology. The processed data used to generate the plots in our manuscript are available via FigShare at https://doi.org/10.6084/m9.figshare.26936107.v2.

The following dataset was generated:

| Author(s) | Year | Dataset title | Dataset URL | Database and Identifier |
|---|---|---|---|---|
| Pham TMQ, Nguyen TN, Nguyen BQT, Tran TPD, Pham NMD, Nguyen HTP, Ho TKC , Nguyen DVL, Nguyen HT, Tran DH, Tran TS, Pham TVN, Le MT, Nguyen TTV, Phan MD, Giang H, Nguyen HN, Tran LS | 2024 | The T Cell Receptor β Chain Repertoire of Tumor Infiltrating Lymphocytes Improves Neoantigen Prediction and Prioritization | https://doi.org/10.6084/m9.figshare.26936107.v3 | figshare, 10.6084/m9.figshare.26936107.v3 |

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
