## [Editor Report · eLife assessment]

The study presents a potentially **valuable** approach by combining two measurements (pHLA binding and pHLA-TCR binding) to improve predictions of which mutations in colorectal cancer are likely to be presented to and recognised by the immune system. While this approach is promising, the evidence supporting the primary claim remains somewhat **incomplete**. The experimental validation of the computational predictions with actual immune responses is still limited, despite the increase in sample size from 4 to 8 in this revision.

---

## [Referee Report · Reviewer #2 (Public Review)]

Summary:

This paper introduces a novel approach for improving personalized cancer immunotherapy by integrating TCR profiling with traditional pHLA binding predictions, addressing the need for more precise neoantigen CRC patients. By analyzing TCR repertoires from tumor-infiltrating lymphocytes and applying machine learning algorithms, the authors developed a predictive model that outperforms conventional methods in specificity and sensitivity. The validation of the model through ELISpot assays confirmed its potential in identifying more effective neoantigens, highlighting the significance of combining TCR and pHLA data for advancing personalized immunotherapy strategies.

Strengths:

(1) Comprehensive Patient Data Collection: The study meticulously collected and analyzed clinical data from 27 CRC patients, ensuring a robust foundation for research findings. The detailed documentation of patient demographics, cancer stages, and pathology information enhances the study's credibility and potential applicability to broader patient populations.

(2) The use of machine learning classifiers (RF, LR, XGB) and the combination of pHLA and pHLA-TCR binding predictions significantly enhance the model's accuracy in identifying immunogenic neoantigens, as evidenced by the high AUC values and improved sensitivity, NPV, and PPV.

(3) The use of experimental validation through ELISpot assays adds a practical dimension to the study, confirming the computational predictions with actual immune responses. The calculation of ranking coverage scores and the comparative analysis between the combined model and the conventional NetMHCpan method demonstrate the superior performance of the combined approach in accurately ranking immunogenic neoantigens.

(4) The use of experimental validation through ELISpot assays adds a practical dimension to the study, confirming the computational predictions with actual immune responses.

Weakness:

The authors have made comprehensive revisions to the original version of the article, and this version has now addressed my concerns.

---

## [Author Response]

The following is the authors’ response to the original reviews.

**Public Reviews:**

**Reviewer #1 (Public Review):**
Summary:This paper reports a number of somewhat disparate findings on a set of colorectal tumour and infiltrating T-cells. The main finding is a combined machine-learning tool which combines two previous state-of-the-art tools, MHC prediction, and T-cell binding prediction to predict immunogenicity. This is then applied to a small set of neoantigens and there is a small-scale validation of the prediciton at the end.Strengths:The prediction of immunogenic neoepitopes is an important and unresolved question.Weaknesses:The paper contains a lot of extraneous material not relevant to the main claim. Conversely, it lacks important detail on the major claim.(1) The analysis of T cell repertoire in Figure 2 seems irrelevant to the rest of the paper. As far as I could ascertain, this data is not used further.

We appreciate the reviewer for their valuable feedback. We concur with the reviewer's observation that the analysis of the TCR repertoire in Figure 2 should be moved to the supplementary section. We have moved Figures 2B to 2F to Supplementary Figure 2.

However, the analysis of TCR profiles is still presented in Figure 2, as it plays a pivotal role in the process of neoantigen selection. This is because the TCR profiles of eight (out of 28) patients were used for neoantigen prediction. We have added the following sentences to the results section to explain the importance of TCR profiling: “Furthermore, characterizing T cell receptors (TCRs) can complement efforts to predict immunogenicity.” (Results, Lines 311-312, Page 11)

(2) The key claim of the paper rests on the performance of the ML algorithm combining NETMHC and pmtNET. In turn, this depends on the selection of peptides for training. I am unclear about how the negative peptides were selected. Are they peptides from the same databases as immunogenic petpides but randomised for MHC? It seems as though there will be a lot of overlap between the peptides used for testing the combined algorithm, and the peptides used for training MHCNet and pmtMHC. If this is so, and depending on the choice of negative peptides, it is surely expected that the tools perform better on immunogenic than on non-immunogenic peptides in Figure 3. I don't fully understand panel G, but there seems very little difference between the TCR ranking and the combined. Why does including the TCR ranking have such a deleterious effect on sensitivity?

We thank the reviewer for their valuable feedback. We believe the reviewer implies 'MHCNet' as NetMHCpan and 'pmtMHC' as pMTnet tools. First, the negative peptides, which have been excluded from PRIME (1), were not randomized with MHC (HLA-I) but were randomized with TCR only. Secondly, the positive peptides selected for our combined algorithms are chosen from many databases such as 10X Genomics, McPAS, VDJdb, IEDB, and TBAdb, while MHCNet uses peptides from the IEDB database and pMTNet uses a totally different dataset from ours for training. Therefore, there is not much overlap between our training data and the training datasets for MHCNet and pMTNet. Thus, the better performance of our tool is not due to overlapping training datasets with these tools or the selection of negative peptides.

To enhance the clarity of the dataset construction, we have added Supplementary Figure 1, which demonstrates the workflow of peptide collection and the random splitting of data to generate the discovery and validation datasets. Additionally, we have revised the following sentence: "To objectively train and evaluate the model, we separated the dataset mentioned above into two subsets: a discovery dataset (70%) and a validation dataset (30%). These subsets are mutually exclusive and do not overlap.” (Methods, lines 221-223, page 8).

Initially, the "combine" label in Figure 3G was confusing and potentially misleading when compared to our subsequent approach using a combined machine learning model. In Figure 3G, the "combine" approach simply aggregates the pHLA and pHLA-TCR criteria, whereas our combined machine learning model employs a more sophisticated algorithm to integrate these criteria effectively. The combined analysis in Figure 3G utilizes a basic "AND" algorithm between pHLA and pHLA-TCR criteria, aiming for high sensitivity in HLA binding and high specificity. However, this approach demonstrated lower efficacy in practice, underscoring the necessity for a more refined integration method through machine learning. This was the key point we intended to convey with Figure 3G. To address this issue, we have revised Figure 3G to replace "combined" with "HLA percentile & TCR ranking" to clarify its purpose and minimize confusion.

(3) The key validation of the model is Figure 5. In 4 patients, the authors report that 6 out 21 neo-antigen peptides give interferon responses > 2 fold above background. Using NETMHC alone (I presume the tool was used to rank peptides according to binding to the respective HLAs in each individual, but this is not clear), identified 2; using the combined tool identified 4. I don't think this is significant by any measure. I don't understand the score shown in panel E but I don't think it alters the underlying statistic.

Acknowledging the limitations of our study's sample size, we proceeded to further validate our findings with four additional patients to acquire more data. The final results revealed that our combined model identified seven peptides eliciting interferon responses greater than a two-fold increase, compared to only three peptides identified by NetMHCpan (Figure 5)

In conclusion, the paper demonstrates that combining MHCNET and pmtMHC results in a modest increase in the ability to discriminate 'immunogenic' from 'non-immunogenic' peptide; however, the strength of this claim is difficult to evaluate without more knowledge about the negative peptides. The experimental validation of this approach in the context of CRC is not convincing.
**Reviewer #2 (Public Review):**
Summary:This paper introduces a novel approach for improving personalized cancer immunotherapy by integrating TCR profiling with traditional pHLA binding predictions, addressing the need for more precise neoantigen CRC patients. By analyzing TCR repertoires from tumor-infiltrating lymphocytes and applying machine learning algorithms, the authors developed a predictive model that outperforms conventional methods in specificity and sensitivity. The validation of the model through ELISpot assays confirmed its potential in identifying more effective neoantigens, highlighting the significance of combining TCR and pHLA data for advancing personalized immunotherapy strategies.Strengths:(1) Comprehensive Patient Data Collection: The study meticulously collected and analyzed clinical data from 27 CRC patients, ensuring a robust foundation for research findings. The detailed documentation of patient demographics, cancer stages, and pathology information enhances the study's credibility and potential applicability to broader patient populations.(2) The use of machine learning classifiers (RF, LR, XGB) and the combination of pHLA and pHLA-TCR binding predictions significantly enhance the model's accuracy in identifying immunogenic neoantigens, as evidenced by the high AUC values and improved sensitivity, NPV, and PPV.(3) The use of experimental validation through ELISpot assays adds a practical dimension to the study, confirming the computational predictions with actual immune responses. The calculation of ranking coverage scores and the comparative analysis between the combined model and the conventional NetMHCpan method demonstrate the superior performance of the combined approach in accurately ranking immunogenic neoantigens.(4) The use of experimental validation through ELISpot assays adds a practical dimension to the study, confirming the computational predictions with actual immune responses.Weaknesses:(1) While multiple advanced tools and algorithms are used, the study could benefit from a more detailed explanation of the rationale behind algorithm choice and parameter settings, ensuring reproducibility and transparency.

We thank the reviewer for their comment. We have revised the explanation regarding the rationale behind algorithm choice and parameter settings as follows: “We examined three machine learning algorithms - Logistic Regression (LR), Random Forest (RF), and Extreme Gradient Boosting (XGB) - for each feature type (pHLA binding, pHLA-TCR binding), as well as for combined features. Feature selection was tested using a k-fold cross-validation approach on the discovery dataset with 'k' set to 10-fold. This process splits the discovery dataset into 10 equal-sized folds, iteratively using 9 folds for training and 1 fold for validation. Model performance was evaluated using the ‘roc_auc’ (Receiver Operating Characteristic Area Under the Curve) metric, which measures the model's ability to distinguish between positive and negative peptides. The average of these scores provides a robust estimate of the model's performance and generalizability. The model with the highest ‘roc_auc’ average score, XGB, was chosen for all features.” (Method, lines 225-234, page 8).

(2) While pHLA-TCR binding displayed higher specificity, its lower sensitivity compared to pHLA binding suggests a trade-off between the two measures. Optimizing the balance between sensitivity and specificity could be crucial for the practical application of these predictions in clinical settings.

We appreciate the reviewer's suggestion. Due to the limited availability of patient blood samples and time constraints for validation, we have chosen to prioritize high specificity and positive predictive value to enhance the selection of neoantigens.

(3) The experimental validation was performed on a limited number of patients (four), which might affect the generalizability of the findings. Increasing the number of patients for validation could provide a more comprehensive assessment of the model's performance.

This has been addressed earlier. Here, we restate it as follows: Acknowledging the limitations of our study's sample size, we proceeded to further validate our findings with four additional patients to acquire more data. The final results revealed that our combined model identified seven peptides eliciting interferon responses greater than a two-fold increase, compared to only three peptides identified by NetMHCpan (Figure 5).

**Reviewer #3 (Public Review):**
Summary:This study presents a new approach of combining two measurements (pHLA binding and pHLA-TCR binding) in order to refine predictions of which patient mutations are likely presented to and recognized by the immune system. Improving such predictions would play an important role in making personalized anti-cancer vaccinations more effective.Strengths:The study combines data from pre-existing tools pVACseq and pMTNet and applies them to a CRC patient population, which the authors show may improve the chance of identifying immunogenic, cancer-derived neoepitopes. Making the datasets collected publicly available would expand beyond the current datasets that typically describe caucasian patients.Weaknesses:It is unclear whether the pNetMHCpan and pMTNet tools used by the authors are entirely independent, as they appear to have been trained on overlapping datasets, which may explain their similar scores. The pHLA-TCR score seems to be driving the effects, but this not discussed in detail.

The HLA percentile from NetMHCpan and the TCR ranking from pMTNet are independent. NetMHCpan predicts the interaction between peptides and MHC class I, while pMTNet predicts the TCR binding specificity of class I MHCs and peptides.Additionally, we partitioned the dataset mentioned above into two subsets: a discovery dataset (70%) and a validation dataset (30%), ensuring no overlap between the training and testing datasets.

To enhance the clarity of the dataset construction, we have added Supplementary Figure 1, which demonstrates the workflow of peptide collection and the random splitting of data to generate the discovery and validation datasets. Additionally, we have revised the following sentence: "To objectively train and evaluate the model, we separated the dataset mentioned above into two subsets: a discovery dataset (70%) and a validation dataset (30%). These subsets are mutually exclusive and do not overlap.” (Methods, lines 221-223, page 8). We also included the dataset construction workflow in Supplementary Figure 1.

Due to sample constraints, the authors were only able to do a limited amount of experimental validation to support their model; this raises questions as to how generalizable the presented results are. It would be desirable to use statistical thresholds to justify cutoffs in ELISPOT data.

We chose a cutoff of 2 for ELISPOT, following the recommendation of the study by Moodie et al. (2). The study provides standardized cutoffs for defining positive responses in ELISPOT assays. It presents revised criteria based on a comprehensive analysis of data from multiple studies, aiming to improve the precision and consistency of immune response measurements across various applications.

Some of the TCR repertoire metrics presented in Figure 2 are incorrectly described as independent variables and do not meaningfully contribute to the paper. The TCR repertoires may have benefitted from deeper sequencing coverage, as many TCRs appear to be supported only by a single read.

We appreciate the reviewer’s feedback. We have moved Figures 2B through 2F to Supplementary Figure 2. We agree with the reviewer that deeper sequencing coverage could potentially benefit the repertoires. However, based on our current sequencing depth, we have observed that many of our samples (14 out of 28) have reached sufficient saturation, as indicated by Figure 2C. The TCR clones selected in our studies are unique molecular identifier (UMI)-collapsed reads, each representing at least three raw reads sharing the same UMI. This approach ensures that the data is robust despite the variability. It is important to note that Tumor-Infiltrating Lymphocytes (TILs) differ across samples, resulting in non-uniform sequencing coverage among them.

**Recommendations for the authors**:
**Reviewer #2 (Recommendations For The Authors):**
(1) Please open source the raw and processed data, code, and software output (NetMHCpan, pMTnet), which are important to verify the results.

NetMHCpan and pMTNet are publicly available software tools (3, 4). In our GitHub repository, we have included links to the GitHub repositories for NetMHCpan and pMTNet (https://github.com/QuynhPham1220/Combined-model).

(2) Comparison with more state-of-the-art neoantigen prediction models could provide a more comprehensive view of the combined model's performance relative to the current field.

To further evaluate our model, we gathered additional public data and assessed its effectiveness in comparison to other models. We utilized immunogenic peptides from databases such as NEPdb (5), NeoPeptide (6), dbPepneo (7), Tantigen (8), and TSNAdb (9), ensuring there was no overlap with the datasets used for training and validation. For non-immunogenic peptides, we used data from 10X Genomics Chromium Single Cell Immune Profiling (10-13).The findings indicate that the combined model from pMTNet and NetMHCpan outperforms NetTCR tool (14). To address the reviewer's inquiry, we have incorporated these results in Supplementary Table 6.

(3) While the combined model shows a positive overall rank coverage score, indicating improved ranking accuracy, the scores are relatively low. Further refinement of the model or the inclusion of additional predictive features might enhance the ranking accuracy.

We appreciate the reviewer’s suggestion. The RankCoverageScore provides an objective evaluation of the rank results derived from the final peptide list generated by the two tools. The combined model achieved a higher RankCoverageScore than pMTNet, indicating its superior ability to identify immunogenic peptides compared to existing in silico tools. In order to provide a more comprehensive assessment, we included an additional four validated samples to recalculate the rank coverage score. The results demonstrate a notable difference between NetMHCpan and the Combined model (-0.37 and 0.04, respectively). We have incorporated these findings into Supplementary Figure 6 to address the reviewer's question. Additionally, we have modified Figure 5E to present a simplified demonstration of the superior performance of the combined model compared to NetMHCpan.

(4) Collect more public data and fine-tune the model. Then you will get a SOTA model for neoantigen selection. I strongly recommend you write Python scripts and open source.

We thank the reviewer for their feedback. We have made the raw and processed data, as well as the model, available on GitHub. Additionally, we have gathered more public data and conducted evaluations to assess its efficiency compared to other methods. You can find the repository here: https://github.com/QuynhPham1220/Combined-model.

**Reviewer #3 (Recommendations For The Authors):**
The Methods section seems good, though HLA calling is more accurate using arcasHLA than OptiType. This would be difficult to correct as OptiType is integrated into pVACtools.

We chose Optitype for its exceptional accuracy, surpassing 99%, in identifying HLA-I alleles from RNA-Seq data. This decision was informed by a recent extensive benchmarking study that evaluated its performance against "gold-standard" HLA genotyping data, as described in the study by Li et al.(15). Furthermore, we have tested two tools using the same RNA-Seq data from FFPE samples. The allele calling accuracy of Optitype was found to be superior to that of Acras-HLA. To address the reviewer's question, we have included these results in Supplementary Table 2, along with the reference to this decision (Method, line 200, page 07).

I am not sufficiently expert in machine learning to assess this part of the methods.TCR beta repertoire analysis of biopsy is highly variable; though my expertise lies largely in sequencing using the 10X genomics platform, typically one sees multiple RNAs per cell. Seeing the majority of TCRs supported by only a single read suggests either problems with RNA capture (particularly in this case where the recovered RNA was split to allow both RNAseq and targeted TCR seq) or that the TCR library was not sequenced deeply enough. I'd like to have seen rarefaction plots of TCR repertoire diversity vs the number of reads to ensure that sufficiently deep sequencing was performed.

We appreciate the suggestions provided by the reviewer. We agree that deeper sequencing coverage could potentially benefit the repertoires. However, based on our current sequencing depth, we have observed that many of our samples (14 out of 28) have reached sufficient saturation, as indicated by Figure 2C. In addition, the TCR clones selected in our studies are unique molecular identifier (UMI)-collapsed reads, each representing at least three raw reads sharing the same UMI. This approach ensures that the data is robust despite variability. It is important to note that Tumor-Infiltrating Lymphocytes (TILs) differ across samples, resulting in non-uniform sequencing coverage among them. We have already added the rarefaction plots of TCR repertoire diversity versus the number of reads in Figure 2C. These have been added to the main text (lines 329-335).

In order to support the authors' conclusions that MSI-H tumors have fewer TCR clonotypes than MSS tumors (Figure S2a) I would have liked to see Figure 2a annotated so that it was easy to distinguish which patient was in which group, as well as the rarefaction plots suggested above, to be sure that the difference represented a real difference between samples and not technical variance (which might occur due to only 4 samples being in the MSI-H group).

We thank the reviewer for their recommendation. Indeed, it's worth noting that the number of MSI-H tumors is fewer than the MSS groups, which is consistent with the distribution observed in colorectal cancer, typically around 15%. This distribution pattern aligns with findings from several previous studies, as highlighted in these studies (16, 17). To provide further clarification on this point, we have included rarefaction plots illustrating TCR repertoire diversity versus the number of reads in Supplementary Figure 3 (line 339). Additionally, MSI-H and MSS samples have been appropriately labeled for clarity.

The authors write: "in accordance with prior investigations, we identified an inverse relationship between TCR clonality and the Shannon index (Supplementary Figure S1)" >> Shannon index is measure of TCR clonality, not an independent variable. The authors may have meant TCR repertoire richness (the absolute number of TCRs), and the Shannon index (a measure of how many unique TCRs are present in the index).

We thank the reviewer for their comment regarding the correlation between the number of TCRs and the Shannon index. We have revised the figure to illustrate the relationship between the number of TCRs and the Shannon index, and we have relocated it to Figure 2B.

The authors continue: "As anticipated, we identified only 58 distinct V (Figure 2C) and 13 distinct J segments (Figure 2D), that collectively generated 184,396 clones across the 27 tumor tissue samples, underscoring the conservation of these segments (Figure 2C & D)" >> it is not clear to me what point the authors are making: it is well known that TCR V and J genes are largely shared between Caucasian populations (https://pubmed.ncbi.nlm.nih.gov/10810226/), and though IMGT lists additional forms of these genes, many are quite rare and are typically not included in the reference sequences used by repertoire analysis software. I would clarify the language in this section to avoid the impression that patient repertoires are only using a restricted set of J genes.

We thank for the reviewer’s feedback. We have revised the sentence as follows: " As anticipated, we identified 59 distinct V segments (Supplementary Figure 2C) and 13 distinct J segments (Supplementary Figure 2D), collectively sharing 185,627 clones across the 28 tumor tissue samples. This underscores the conservation of these segments (Supplementary Figure 2C & D)” (Result, lines 354-356, page 12)

As a result I would suggest moving Figure 2 with the exception of 2A into the supplementals - I would have been more interested in a plot showing the distribution of TCRs by frequency, i.e. how what proportion of clones are hyperexpanded, moderately expanded etc. This would be a better measure of the likely immune responses.

We thank the reviewer for their comment. With the exception of Figure 2A, we have relocated Figures 2B through 2F to Supplementary Figure 2.

The authors write "To accomplish this, we gathered HLA and TCRβ sequences from established datasets containing immunogenic and non-immunogenic peptides (Supplementary Table 3)" >> The authors mean to refer to Table S4.

We appreciate the reviewer's feedback. Here's the revised sentence: "To accomplish this, we gathered HLA and TCRβ sequences from established datasets containing immunogenic and non-immunogenic pHLA-TCR complexes (Supplementary Table 5)” (lines 368-370).

The authors write "As anticipated, our analysis revealed a significantly higher prevalence of peptides with robust HLA binding (percentile rank < 2%) among immunogenic peptides in contrast to their non-immunogenic counterparts (Figure 3A & B, p< 0.00001)" >> this is not surprising, as tools such as NetMHCpan are trained on databases of immunogenic peptides, and thus it is likely that these aren't independent measures (in https://academic.oup.com/nar/article/48/W1/W449/5837056 the authors state that "The training data have been vastly extended by accumulating MHC BA and EL data from the public domain. In particular, EL data were extended to include MA data"). In the pMTNet paper it is stated that pMNet encoded pMHC information using "the exact data that were used to train the netMHCpan model" >> While I am not sufficiently expert to review details on machine learning training models, it would seem that the pHLA scores from NetMHCpan and pMTNet may not be independent, which would explain the concordance in scores that the authors describe in Figures 3B and 3D. I would invite the authors to comment on this.

The HLA percentiles from NetMHCpan and TCR rankings from pMTNet are independent. NetMHCpan predicts the interaction between peptides and MHC class I, while pMTNet predicts the TCR binding specificity of class I MHCs and peptides. NetMHCpan is trained to predict peptide-MHC class I interactions by integrating binding affinity and MS eluted ligand data, using a second output neuron in the NNAlign approach. This setup produces scores for both binding affinity and ligand elution. In contrast, pMTNet predicts TCR binding specificity of class I pMHCs through three steps:

(1) Training a numeric embedding of pMHCs (class I only) to numerically represent protein sequences of antigens and MHCs.

(2) Training an embedding of TCR sequences using stacked auto-encoders to numerically encode TCR sequence text strings.

(3) Creating a deep neural network combining these two embeddings to integrate knowledge from TCRs, antigenic peptide sequences, and MHC alleles. Fine-tuning is employed to finalize the prediction model for TCR-pMHC pairing.

Therefore, pHLA scores from NetMHCpan and pMTNet are independent. Furthermore, Figures 3B and 3D do not show concordance in scores, as there was no equivalence in the percentage of immunogenic and non-immunogenic peptides in the two groups (≥2 HLA percentile and ≥2 TCR percentile).

Many of the authors of this paper were also authors of the epiTCR paper, would this not have been a better choice of tool for assessing pHLA-TCR binding than pMTNet?

When we started this project, EpiTCR had not been completed. Therefore, we chose pMTNet, which had demonstrated good performance and high accuracy at that time. The validated performance of EpiTCR is an ongoing project that will implement immunogenic assays (ELISpot and single-cell sequencing) to assess the prediction and ranking of neoantigens. This study is also mentioned in the discussion: "Moreover, to improve the accuracy and effectiveness of the machine learning model in predicting and ranking neoantigens, we have developed an in-house tool called EpiTCR. This tool will utilize immunogenic assays, such as ELISpot and single-cell sequencing, for validation." (lines 532-535).

In Figure 3G it would appear that the pHLA-TCR score is driving the interaction, could the authors comment on this?

The authors sincerely appreciate the reviewer for their valuable feedback. Initially, the "combine" label in Figure 3G was confusing and potentially misleading when compared to our subsequent approach using a combined machine learning model. In Figure 3G, the "combine" approach simply aggregates the pHLA and pHLA-TCR criteria, whereas our combined machine learning model employs a more sophisticated algorithm to integrate these criteria effectively.

The combined analysis in Figure 3G utilizes a basic "AND" algorithm between pHLA and pHLA-TCR criteria, aiming for high sensitivity in HLA binding and high specificity. However, this approach demonstrated lower efficacy in practice, underscoring the necessity for a more refined integration method through machine learning. This was the key point we intended to convey with Figure 3G. To address this issue, we have revised Figure 3G to replace "combined" with "HLA percentile & TCR ranking" to clarify its purpose and minimize confusion.

In Figure 4A I would invite the authors to comment on how they chose the sample sizes they did for the discovery and validation datasets: the numbers seem rather random. I would question whether a training dataset in which 20% of the peptides are immunogenic accurately represents the case in patients, where I believe immunogenic peptides are less frequent (as in Figure 5).

We aimed to maximize the number of experimentally validated immunogenic peptides, including those from viruses, with only a small percentage from tumors available for training. This limitation is inherent in the field. However, our ultimate objective is to develop a tool capable of accurately predicting peptide immunogenicity irrespective of their source. Therefore, the current percentage of immunogenic peptides may not accurately reflect real-world patient cases, but this is not crucial to our development goals.

For Figure 5C I would invite the authors to consider adding a statistical test to justify the cutoff at 2fold enrichments.

Thank you for your feedback. Instead of conducting a statistical test, we have implemented standardized cutoffs as defined in the cited study (2). This research introduces refined criteria for identifying positive responses in ELISPOT assays through a comprehensive analysis of data from multiple studies. These criteria aim to improve the accuracy and consistency of immune response measurements across various applications. The reference to this study has been properly incorporated into the manuscript (Method, line 281, page 10).

Minor points:"paired white blood cells" >> use "paired Peripheral Blood Mononuclear Cells".

We appreciate the reviewer for the feedback. We agree with the reviewer's observation. The sentence has been revised as follows: "Initially, DNA sequencing of tumor tissues and paired Peripheral Blood Mononuclear Cells identifies cancer-associated genomic mutations. RNA sequencing then determines the patient's HLA-I allele profile and the gene expression levels of mutated genes." (Introduction, lines 55-58, page 2).

"while RNA sequencing determines the patient's HLA-I allele profile and gene expression levels of mutated genes." >> RNA sequencing covers both the mutant and reference form of the gene, allowing assessment of variant allele frequency."the current approach's impact on patient outcomes remains limited due to the scarcity of effective immunogenic neoantigens identified for each patient" >> Some clearer language here would have been preferred as different tumor types have different mutational loads

We thank the reviewer for their valuable feedback. We agree with the reviewer's observation. The passage has been revised accordingly: “The current approach's impact on patient outcomes remains limited due to the scarcity of mutations in cancer patients that lead to effective immunogenic neoantigens.” (Introduction, lines 62-64, page 3).

References

(1) J. Schmidt *et al.*, Prediction of neo-epitope immunogenicity reveals TCR recognition determinants and provides insight into immunoediting. *Cell Rep Med*
**2**, 100194 (2021).

(2) Z. Moodie *et al.*, Response definition criteria for ELISPOT assays revisited. *Cancer Immunol Immunother*
**59**, 1489-1501 (2010).

(3) V. Jurtz *et al.*, NetMHCpan-4.0: Improved Peptide-MHC Class I Interaction Predictions Integrating Eluted Ligand and Peptide Binding Affinity Data. *J Immunol*
**199**, 3360-3368 (2017).

(4) T. Lu *et al.*, Deep learning-based prediction of the T cell receptor-antigen binding specificity. *Nat Mach Intell*
**3**, 864-875 (2021).

(5) J. Xia *et al.*, NEPdb: A Database of T-Cell Experimentally-Validated Neoantigens and Pan-Cancer Predicted Neoepitopes for Cancer Immunotherapy. *Front Immunol*
**12**, 644637 (2021).

(6) W. J. Zhou *et al.*, NeoPeptide: an immunoinformatic database of T-cell-defined neoantigens. *Database (Oxford)*
**2019** (2019).

(7) X. Tan *et al.*, dbPepNeo: a manually curated database for human tumor neoantigen peptides. *Database (Oxford)*
**2020** (2020).

(8) G. Zhang, L. Chitkushev, L. R. Olsen, D. B. Keskin, V. Brusic, TANTIGEN 2.0: a knowledge base of tumor T cell antigens and epitopes. *BMC Bioinformatics*
**22**, 40 (2021).

(9) J. Wu *et al.*, TSNAdb: A Database for Tumor-specific Neoantigens from Immunogenomics Data Analysis. *Genomics Proteomics Bioinformatics*
**16**, 276-282 (2018).

(10) https://www.10xgenomics.com/resources/datasets/cd-8-plus-t-cells-of-healthy-donor-1-1-standard-3-0-2.

(11) https://www.10xgenomics.com/resources/datasets/cd-8-plus-t-cells-of-healthy-donor-2-1-standard-3-0-2.

(12) https://www.10xgenomics.com/resources/datasets/cd-8-plus-t-cells-of-healthy-donor-3-1-standard-3-0-2.

(13) https://www.10xgenomics.com/resources/datasets/cd-8-plus-t-cells-of-healthy-donor-4-1-standard-3-0-2.

(14) A. Montemurro *et al.*, NetTCR-2.0 enables accurate prediction of TCR-peptide binding by using paired TCRalpha and beta sequence data. *Commun Biol*
**4**, 1060 (2021).

(15) G. Li *et al.*, Splicing neoantigen discovery with SNAF reveals shared targets for cancer immunotherapy. *Sci Transl Med*
**16**, eade2886 (2024).

(16) Z. Gatalica, S. Vranic, J. Xiu, J. Swensen, S. Reddy, High microsatellite instability (MSI-H) colorectal carcinoma: a brief review of predictive biomarkers in the era of personalized medicine. *Fam Cancer*
**15**, 405-412 (2016).

(17) N. Mulet-Margalef *et al.*, Challenges and Therapeutic Opportunities in the dMMR/MSI-H Colorectal Cancer Landscape. *Cancers (Basel)*
**15** (2023).